# Ubiquitin Engineering for Interrogating the Ubiquitin–Proteasome System and Novel Therapeutic Strategies

**DOI:** 10.3390/cells12162117

**Published:** 2023-08-21

**Authors:** Jason Q. Tang, Mary M. Marchand, Gianluca Veggiani

**Affiliations:** 1Donnelly Centre for Cellular and Biomolecular Research, University of Toronto, 160 College Street, Toronto, ON M5S3E1, Canada; 2Department of Molecular Genetics, University of Toronto, 160 College Street, Toronto, ON M5S3E1, Canada; 3Department of Pathobiological Sciences, School of Veterinary Medicine, Louisiana State University, Baton Rouge, LA 70803, USA; 4Division of Biotechnology and Molecular Medicine, Louisiana State University, Baton Rouge, LA 70803, USA

**Keywords:** ubiquitin, ubiquitin variants (UbVs), ubiquitin–proteasome system (UPS), deubiquitinating enzymes (DUBs), degradation

## Abstract

Protein turnover, a highly regulated process governed by the ubiquitin–proteasome system (UPS), is essential for maintaining cellular homeostasis. Dysregulation of the UPS has been implicated in various diseases, including viral infections and cancer, making the proteins in the UPS attractive targets for therapeutic intervention. However, the functional and structural redundancies of UPS enzymes present challenges in identifying precise drug targets and achieving target selectivity. Consequently, only 26S proteasome inhibitors have successfully advanced to clinical use thus far. To overcome these obstacles, engineered peptides and proteins, particularly engineered ubiquitin, have emerged as promising alternatives. In this review, we examine the impact of engineered ubiquitin on UPS and non-UPS proteins, as well as on viral enzymes. Furthermore, we explore their potential to guide the development of small molecules targeting novel surfaces, thereby expanding the range of druggable targets.

## 1. Introduction

Precise protein turnover is essential for maintaining cell homeostasis, and its dysregulation is involved in numerous pathologies [1]. In mammalian cells, the ubiquitin–proteasome system (UPS) is the main pathway for protein degradation [2] and requires the coordinated activity of an E1–E2–E3 enzyme cascade and subsequent covalent conjugation of ubiquitin (Ub) to target proteins [3]. In addition to protein degradation, ubiquitination can also alter protein localization, modulate protein activity, or act as a scaffold for the recruitment of signaling proteins [4]. Ub signaling is also implicated in other critical cellular processes such as autophagy, mitophagy, cell-cycle control, metabolic pathways, DNA stability, repair, and replication [4].

The ubiquitination process begins with the activation of Ub via an E1 enzyme (ubiquitin-activating enzyme) upon which the activated Ub is transferred to an E2 enzyme (ubiquitin-conjugating enzyme) which interacts with an E3 ubiquitin ligase (Figure 1a). E3 ligases are responsible for recognizing specific target proteins and mediating the covalent linkage of Ub by the formation of an isopeptide bond to the lysine residues of substrate proteins (Figure 1a). The seven lysine residues on Ub (K6, K11, K27, K29, K33, K48, or K63) or the Ub N-terminus (Figure 1b) can serve as target sites for the conjugation of other ubiquitin molecules, resulting in the formation of diverse polyubiquitin chains including homotypic, heterotypic, or branched linkages that greatly expand the complexity of ubiquitin signaling (Figure 1). Depending on their linkages, Ub chains encode specific cellular effects. For example, K48-linked polyubiquitin chains target proteins for degradation by the ubiquitin 26S proteasome, whereas K63-linked polyubiquitin chains are involved in membrane protein recycling and lysosomal degradation. Subsequently, specific Ub signals can be recognized by more than 20 different ubiquitin-binding domain (UBD) families [5,6]. These UBDs, such as the ubiquitin-interacting motifs (UIMs), are often found in tandem repeats and across many different protein families with a wide variety of functions [7,8,9]. Therefore, UBDs translate the ubiquitin code into signaling pathways. Ubiquitin signaling is also constantly regulated by deubiquitinating enzymes (DUBs) that hydrolyze the ubiquitin isopeptide bonds, thereby acting as signal editors and modulators (Figure 1a).

The dysregulation of the UPS has been implicated in the pathogenesis of various diseases, including cancer, neurodegenerative disorders, and autoimmune conditions. For example, in cancer, aberrant UPS activity contributes to the accumulation of oncogenic proteins or the degradation of tumor suppressors (e.g., p53), promoting tumor growth and survival [10,11]. Therefore, targeting specific components of the UPS offers opportunities for therapeutic intervention by restoring protein balance and inhibiting tumor growth. These efforts have primarily been focused on the development of compounds able to modulate enzyme activity in the UPS or hijack Ub-related enzyme activity to induce degradation (e.g., PROTACs, molecular glues) or stabilization of new ligands [12,13,14]. However, developing small molecules targeting enzymes in the UPS presents several challenges. The large functional and structural redundancy of enzymes in the UPS makes it challenging to precisely identify suitable drug targets, achieve target selectivity, and navigate their dynamic nature. These obstacles contribute to the difficulty of translating promising preclinical results into successful therapies that can be used in the clinic. As such, the only small-molecule inhibitors approved by the Food and Drug Administration (FDA), among all UPS components, are those targeting the 26S proteasome [15,16].

Peptides and biologics offer unique advantages over small molecules as they have larger rapidly developable surfaces for interactions allowing for greater binding affinity and specificity [17,18]. However, unique challenges such as their delivery into cells and the specific targeting of UPS proteins still need to be addressed. Furthermore, to be effective in the reducing cytoplasmic environment, peptides and biologics need to be highly stable and capable of being expressed in diverse cellular compartments for broad applicability.

Over the years, several different scaffolds with favorable characteristics have been used to develop probes of intracellular proteins, but they have not been used for systematic targeting of the UPS [19]. Given their intrinsic properties, including their ability to interact with proteins in the UPS, engineered ubiquitin variants (UbVs) have emerged as a powerful means for investigating and developing UPS modulators [20].

Here, we present an overview of the advancements in UbV technology and its applications in developing inhibitors and activators of the UPS. Furthermore, we discuss how these innovative tools are shaping the development of novel therapeutic strategies for various diseases.

## 2. Ubiquitin Engineering

The intrinsic properties of Ub represent an ideal framework for the development of intracellular modulators. Notably, Ub exhibits exceptional thermostability and remains stable across a wide range of pH values [21]. Furthermore, Ub lacks cysteine residues, which enables folding in the reducing environment within cells. Moreover, the ubiquitous expression of Ub in every cellular compartment makes it well suited for targeting proteins with diverse functions and cellular localization. Another key feature of Ub is its ability to engage in low-affinity but specific interactions with many proteins in the UPS. These interactions most commonly occur through an extended Ile44-hydrophobic patch that includes Leu8, Ile44, His68, and Val70 (Figure 1c) [22]. Nevertheless, Ub can also interact with its binding partners through additional regions, including its C-terminus and the surface formed by the α/β groove (Figure 1c), thus expanding the modes of interaction and increasing the diversity of its binding partners [22].

These unique properties, along with Ub’s tolerance to mutations in both surface-exposed and buried residues [23,24], have captured the attention of numerous research groups for the development of UbVs to selectively target specific proteins or protein complexes within the UPS.

Several established methods have been developed for engineering the repertoire of UbVs. Typically, this begins with a library construction guided by the vast, publicly available structural data of Ub in complex with members of the UPS. Both computational and rational designs have been used to identify residues that can be mutated to improve UbV binding. Subsequently, specific amino acids in the Ub sequence are diversified using mutagenesis methods to generate large libraries encoding a diverse set of UbVs (Figure 2). These libraries (phage-displayed, yeast-displayed, and yeast-two-hybrid) are then subjected to affinity selections, where a target of interest is used as bait for isolating UbVs that bind specifically to it (Figure 2). Through iterative rounds of affinity selection, UbVs with an enhanced affinity for the target can be obtained (Figure 2, right panel). In addition to enhancing affinity, fine-tuning the binding selectivity of UbVs is crucial for their specificity over different homologous proteins.

To achieve this, UbV libraries are typically depleted of binders that recognize protein homologs before performing affinity selections against the desired target protein. Additional strategies such as second-generation libraries based on previously isolated UbVs are also used. These strategic approaches have facilitated the generation of diverse UbVs, enabling the exploration of different binding interfaces, yielding high-affinity and specific probes for a wide range of UPS targets.

## 3. Targeting Enzymes of the Ubiquitin–Proteasome System

### 3.1. UbVs Targeting E2 Ubiquitin-Conjugating Enzymes

Ubiquitin-conjugating E2 enzymes are a central component of ubiquitination, and their role as protein regulators outside the traditional Ub transfer pathways, as well as in human pathologies, is emerging [25,26,27]. Therefore, UbV probes have the potential of elucidating the molecular function of E2 enzymes and discovering new therapeutic strategies.

To this end, UbVs have been recently developed for specific interactions with the E2 Ube2k, whose aberrant expression contributes to multiple neurodegenerative diseases including Parkinson’s and Alzheimer’s disease [28,29,30]. Using phage display, Middleton et al. discovered potent and specific UbV inhibitors of Ube2k-mediated Ub transfer [31]. Structural studies revealed that these UbVs block the engagement of both the E1 enzyme and the E3 ligase [31]. Remarkably, this effect did not occur via interactions with the Ube2k catalytic site, whose conservation across E2 enzymes prevented the development of selective inhibitors.

In several E2s, noncovalent Ub binding to the backside of the enzyme, opposite to the catalytic site, has been shown to regulate their activity [32]. By exploiting this property, Garg et al. developed UbVs specifically interacting with this surface, resulting in the modulation of the enzymatic activity of Ube2D1 (Figure 3a), Ube2V1, and Ube2G1 [33]. While E2-specific UbVs did not affect Ub charging in Ube2D1 and Ube2V1, UbVs reduced processive Ub chain formation by Ube2D1 and substantially reduced Ube2V1-mediated di-Ub formation [33]. In Ube2G1, instead, UbVs inhibited E1-mediated Ub charging. In all cases, UbVs did not completely abrogate E2 activity but rather modulated their enzymatic activity [33].

So far, UbVs targeting E2 enzymes have shown their potential for unveiling the molecular mechanism of E2-mediated E3 ligase regulation and have shed light on the previously underappreciated backsides of E2s, revealing them as an attractive surface for pharmacological modulation.

### 3.2. UbVs Targeting E3 Ligases

E3 ubiquitin ligases are key components of the UPS, catalyzing the transfer of Ub to substrate proteins. In humans, over 600 E3 ligases are divided into four classes: HECT type, U-box type, RING-finger type, and RBR type [34,35]. These enzymes have emerged as key drug targets for the discovery of inhibitors that can block their dysregulated activity [36] or engagers that specifically recruit E3 ligases for induced ubiquitination and targeted protein degradation [37]. Despite recent successes in drugging E3 ligases, targeting these enzymes with small molecules still poses significant challenges, and alternative methods are desirable. To address this issue, a vast number of UbVs targeting different classes of E3 ligases have been successfully developed.

#### 3.2.1. UbVs for HECT-E3 Ligases

The NEDD4/NEDD4-like subfamily of HECT-E3 ligases is regulated by autoubiquitination and its aberrant expression has been implicated in several diseases [38]. In particular, mutations or deficiency of NEDD4L prevent degradation of the epithelial sodium channel (ENaC), leading to its accumulation and increased sodium reabsorption, resulting in Liddle syndrome [39]. Thus, small molecules able to enhance NEDD4L activity would represent promising therapeutic candidates [39]. By using a Ub phage display library, Zhang and coworkers isolated UbVs specifically targeting with high affinity the HECT domain of NEDD4L among another 20 HECT-type E3 ligases [40]. Notably, a UbV binding the HECT N-lobe exosite of NEDD4L (UbV.NL1) (Figure 3a) resulted in the unexpected activation of the enzyme and enhanced ubiquitination, reducing the luminal swelling within mouse distal colon organoids [40]. Surprisingly, the same UbV significantly reduced cellular migration of colorectal cancer cells, suggesting a potential role of NEDD4L in metastatic phenotypes [40]. Thus, UbV activators of NEDD4L (Figure 3a) could be exploited as novel means for the treatment of hypertension or cancer cell migration, in a context-dependent fashion.

In addition to NEDD4L, other UbV activators and inhibitors have also been characterized for the NEDD4/NEDD4-like subfamily of HECT-E3 ligases including WWP1, WWP2, ITCH, Rsp5, HACE1, and NEDD4, further demonstrating that UbV technology allows the rapid isolation of modulators of E3 ligases [40,41].

#### 3.2.2. UbVs for U-Box E3 Ligases

Many U-box family E3 ligases have been implicated in the quality control of proteins. Among this family, UBE4B is a significant regulator of cancer progression as, via its interaction with p53, it inhibits p53 transactivation and suppression of apoptosis [42]. Therefore, UBE4B plays a crucial role in cancer therapy. Affinity selections against the U-box of UBE4B yielded a UbV (UbV.E4B) that exclusively recognized UBE4B among a panel of other RING and different U-box domains derived from distinct E3 ligases [43]. The high affinity of UbV.E4B (K_D_ 1.9 µM) ensured the potent inhibition of polyubiquitin chain elongation by blocking the E2 binding site, significantly decreasing ubiquitination of p53 in cells.

#### 3.2.3. UbVs for RING-E3 Ligases

UbVs can also be powerful tools for studying the pathogenic role of RING E3 ligases in cancers. For example, oncogenic CBL regulates epidermal growth factor receptor (EGFR) via its ubiquitination, downregulation, and lysosomal degradation [44]. A UbV binding the RING domain of active, phosphorylated CBL and stabilizing EGFR provides a powerful probe for dissecting EGFR trafficking and signaling [43].

Furthermore, by targeting the RING domain of XIAP with a UbV that forms a stable homodimer (Figure 3a), the ligase activity of the enzyme was enhanced, leading to substantial ubiquitination of the XIAP substrate mature SMAC (mSMAC), further demonstrating the binding versatility of UbVs and their potential in dissecting ubiquitination-dependent pathogenic pathways [43].

UbVs have also been instrumental for investigating multicomponent E3 ligase complexes such as the Skp1-Cul1/Rbx-F-box protein complex (SCF). By binding the F-box of Fbw7 in a site which is required for the direct interaction of Skp1–Fbw7 with Cul1 (a member of the Cullin family that provides a scaffold for ubiquitin ligases) [45], Ubv.Fw7.1 potently inhibited the polyubiquitination activity in vitro [46,47]. Furthermore, the intracellular expression of SCF-specific UbVs showed a reduced degradation of the SCF-Fbw7 substrates Cyclin E and c-Myc, as well as of SCF-Fbw11 substrates Cdc25A and Wee1, to levels comparable to those observed upon the expression of siRNAs targeting both SCFs [46]. Additionally, targeting the F-box of the Skp1-Fbl1 and SCF-Fbo11 complexes with UbVs (Figure 3b) resulted in the specific stabilization of p27 and Snail, respectively [47], further confirming that preventing Cul1 binding to SCF E3 ligases successfully inhibits ubiquitination in cells.

Given the expansive impact of UbVs on modulating E3 ligase activity, we foresee advancements in developing direct inhibitors that target pathogenic E3 ligases. For example, the TRAF family of RING-E3 ligases holds significance in cellular invasion and proliferation in breast cancers [48]. Specifically, TRAF6 is prominently associated with the polyubiquitination-stimulated cleavage of TβRI by TACE and PS1, initiating downstream activation of genes linked to cancer invasiveness [49,50].

### 3.3. UbVs Targeting Deubiquitinating Enzymes

Deubiquitinating enzymes (DUBs) are key regulators of Ub signaling through the hydrolysis of Ub and polyubiquitin chains. In humans, there are more than 100 DUBs classified into seven distinct protein families: the ubiquitin-specific proteases (USPs), ovarian tumor proteases (OTUs), Machado–Josephin domain proteases (MJDs), ubiquitin C-terminal hydrolases (UCHs), the JAB1/MPN/MOV34 family (JAMM), the motif interacting with ubiquitin-containing novel DUB (MINDY) family, and the zinc finger-containing ubiquitin peptidase (ZUFSP). The central role of DUBs in maintaining protein homeostasis has led to extensive investigations into their involvement in tumorigenesis and cancer progression [51,52]. For example, certain DUBs, including BAP1, UCHL1, CYLD, and USP22, exhibit intrinsic oncogenic or tumor-suppressor activities [51]. Moreover, DUBs like USP22 play a role in promoting tumor development through the regulation of epigenetic changes [53,54], while others such as USP28 and USP7 are involved in controlling the turnover of oncogenes or tumor suppressors [55,56].

As a result, DUBs have emerged as promising targets for therapeutic intervention [57]. However, the development of inhibitors of the catalytic activity of DUBs has been greatly impeded by challenges associated with their highly conserved active sites [52,58]. These challenges are particularly pronounced in the case of USPs, which form the largest family of DUBs with a repertoire of 56 unique members.

#### 3.3.1. UbV Inhibitors of Ubiquitin-Specific Proteases

Oncogenic USP7 has garnered significant attention in drug discovery campaigns for over a decade. In fact, its deubiquitinating activity plays a key role in stabilizing MDM2, an E3 ligase that targets the tumor suppressor p53 for degradation in numerous cancer types. Tackling the USP7 challenge, Zhang et al. employed RosettaDesign in silico modeling to identify specific Ub conformations that would exhibit strong interactions with the catalytic core of USP7 and design large UbV phage-displayed libraries [59]. From these libraries, Zhang et al. successfully engineered a tight and specific UbV, named U7Ub25, with a remodeled backbone and optimized core packing. Further engineering of solvent-exposed residues led to the development of another UbV, U7Ub25.2540, that displayed a 4-fold enhanced affinity and potently inhibited USP7 in vitro. The intracellular expression of U7Ub25.2540 in cancer cells inhibited endogenous USP7, enhancing MDM2 proteasomal degradation and the stabilization of p53 [59]. Building upon this success, researchers have developed additional UbVs targeting USP7. Through the utilization of a naïve UbV phage display library, they isolated UbV7.2, which exhibited substantially improved binding specificity when compared to U7Ub.2540, along with a remarkable enhancement in the degradation of MDM2 [60].

Additional phage-display-mediated affinity-selection campaigns were focused on the identification of UbVs targeting USP10, a key regulator of p53 that counteracts MDM2-mediated p53 nuclear export and degradation [61]. USP10-specific UbVs exhibited potent inhibitory activity against the enzyme, leading to destabilization of p53 and increased nuclear export of p53 to the cytoplasm, providing yet another strategy for modulation of the p53/MDM2 pathway [60].

Another important player in the regulation of the p53/MDM2 pathway is USP2a. This deubiquitinase is involved in stabilizing MDM2, which subsequently promotes the degradation of p53, making USP2a an alternative target for p53 stabilization [62]. Through screening of a yeast-two-hybrid (Y2H) library obtained via diversification of four positions in the Ub wild-type sequence, two UbVs, namely, UbV2.2 and UbV2.6, were identified as potent inhibitors of USP2 with IC_50_ values in the single-digit nanomolar range [63]. These UbVs displayed exceptional specificity to USP2 when compared to a panel of 35 other DUBs and several E2s when tested by Y2H. Moreover, both UbVs also inhibited MDM2 deubiquitination in a model cell line [63]. Interestingly, a previously selected UbV targeting USP2 reported by Ernst et al. did not exhibit any activity in cells [41], highlighting the potential advantages of using an intracellular directed evolution system for engineering UbVs.

Similarly to what was described for USP7, Sun et al. computationally designed 6000 UbVs and used phage display and Y2H to identify inhibitors of USP21 [64]. This strategy resulted in the identification of numerous UbVs with remarkable potency, requiring low nanomolar concentrations for 50% inhibition of USP21 activity [64]. Strikingly, a machine learning model generated from the identified UbVs correctly identified 92% of USP21-binding variants that were previously identified by Ernst et al. using a naïve UbV phage display library [41]. These findings demonstrate that computational design methods indeed enable the identification of target specific UbVs. We anticipate that the current advancements in machine learning will further aid in the design and optimization of UbVs in future campaigns, enabling the generation of more focused and efficient UbV phage display libraries with desired properties.

UbVs have also been developed for USP15, a pleiotropic DUB with substrates within the TGF-β, NF-κB, and p53/MDM2 pathways, whose activity is relevant in a number of cancers [65,66,67,68]. Initially, UbVs recognizing the catalytic domain of USP15 (Figure 3a) exhibited potent inhibition of its activity [69]. However, these UbVs also demonstrated some degree of inhibition toward its paralog USP4, due to a significant sequence identity (76%) in their core catalytic domains [69]. In an attempt to improve binding specificity, Teyra et al. implemented engineering strategies by extending the β1-β2 and β3-β4 loops of Ub [69]. This approach enhanced the binding affinity but unexpectedly led to reduced specificity for USP15. In fact, structural analysis showed that the extended UbV β3-β4 loop established additional contacts with the catalytic domain of USP15, compromising its selectivity. In addition to targeting the catalytic domain, a UbV recognizing the USP15 noncatalytic DUSP domain exhibited an unexpected β1-strand swap, resulting in the formation of a UbV dimer that provided a novel binding surface (Figure 3b) [69]. Teyra and coworkers also isolated UbVs against additional noncatalytic domains of USP15 (Ubl-1 and Ubl-2) [69]. To further enhance both affinity and specificity through avidity effects, a UbV dimer was created by linking in tandem UbV.15.1 (targeting the enzyme catalytic site) and UbV.15.D (a dimeric UbV specific for the DUSP domain). This fusion, named UbV.15.1/15.D, demonstrated improved specificity and inhibitory activity against USP15 in vitro, while its intracellular expression led to enhanced polyubiquitination of SMURF2 and monoubiquitination of TRIM25 [69].

UbVs have also been developed for modulating the endocytosis of cell receptors. Specifically, USP8 plays a significant role in the regulation of ligand-mediated endocytosis for multiple receptors, including EGFR, whose deubiquitination is crucial for preventing its lysosomal degradation [70]. Ernst et al. isolated a highly specific UbV (Ubv.8.2) which binds the catalytic site of USP8 with a noncanonical orientation [41] (Figure 3a). Nevertheless, Ubv.8.2 effectively blocked the Ub-binding site of the enzyme and strongly inhibited USP8 activity in vitro. Confocal microscopy experiments revealed that Ubv.8.2 expression enhanced the transfer of EGFR from endosomes to lysosomes, as evidenced by increased colocalization with lysosome-associated protein 1 (LAMP1) and decreased colocalization with the endosomal-recycling marker Ras-related protein RAB11. Consequently, in response to EGF, endogenous EGFR levels were downregulated at a faster rate, indicating effective USP8 inhibition [41].

#### 3.3.2. UbV Inhibitors of the JAMM Family

The JAB1/MPN/MOV34 (JAMM) family of DUBs represents the only metalloenzymes within the UPS that rely on the coordination of a zinc ion in their JAMM domains to exert catalytic activity [71]. Several members of the JAMM family have been implicated in various human diseases [71]. Due to their distinct catalytic mechanism and relatively fewer numbers compared to other DUB families, JAMMs have emerged as a novel class of drug targets [71]. One member of this family, STAMBP, is a main contributor in microcephaly–capillary malformation (MIC-CAP) syndrome [72] and has also been implicated in impeding the lysosomal degradation of the proinflammatory inflammasome constituent NALP7 [73]. Consequently, pharmacologically targeting STAMBP holds promising potential for mitigating proinflammatory stress and associated pathologies.

Despite the complete conservation of the catalytic core across all members of the JAMM family, a UbV isolated through phage display exhibited remarkable specificity for the JAMM domain of STAMBP [74]. In K63-diubiquitin and K63-linked polyubiquitin cleavage assays conducted in vitro, this UbV displayed a potent inhibition of STAMBP cleavage, surpassing the inhibitory effect of BC-1471, a known STAMBP ubiquitin-binding site inhibitor [73]. Structural investigations further revealed that the UbV binds to the distal K63-linked ubiquitin-binding site (Figure 3a) [74], highlighting the potential of targeting this surface to impede the processivity of STAMBP rather than the conserved catalytic site.

#### 3.3.3. UbVs as Inhibitors of Viral DUBs

Deubiquitinating enzymes play a key role in infections, as viral DUBs (vDUBs) actively exploit the host UPS to suppress innate antiviral responses and facilitate viral replication [75]. Therefore, there has been significant interest in the development of UbVs capable of inhibiting vDUBs as a potential antiviral strategy.

Using a naïve UbV phage display library, UbVs were first isolated against the USP-like papain-like protease (PL^pro^) domain of the Middle East respiratory syndrome coronavirus (MERS-CoV), as well as against the OTU domain of the chymotrypsin-like protease of the Crimean–Congo hemorrhagic fever virus (CCHFV) [76]. These UbVs demonstrated high binding affinity to their respective vDUBs and exhibited potent inhibitory effects. Specifically, UbV.ME.4 (Figure 3a), targeting MERS-CoV PL^pro^ and UbV.CC.4, binding CCHFV OTU, effectively inhibited deubiquitination and deISGlyation processes [76]. Furthermore, they demonstrated the inhibition of tetra-K48-linked and tetra-K63-linked polyubiquitin processing. Intriguingly, the intracellular expression of UbV.ME.4 resulted in a reduced suppression of the IFN-β promoter and a complete blockade of viral replicative polyprotein processing. As a result, UbV.ME.4 exhibited a robust inhibition of viral replication [76].

In a similar study, Hung et al. utilized molecular dynamics (MD) simulations to examine the interaction between Ub and MERS-CoV PL^pro^ [77]. By employing side-chain dihedral correlation and force distribution analyses, they successfully identified a minimal number of residues that could be selectively mutated to develop potent UbVs targeting MERS-CoV PL^pro^. This approach offered advantages in terms of modifying only a limited set of residues, thus minimizing the potential negative effects on UbV stability. Although the resulting MERS-CoV PL^pro^ inhibitors exhibited weaker IC_50_ values compared to UbVs obtained through phage display, this study highlights the potential of this targeted mutational strategy for the development of effective UbV-based inhibitors [77].

More recently, in response to the COVID-19 pandemic, UbVs have been developed against the SARS-CoV-2 PL^Pro^ [78]. Phage display selections and affinity-maturation campaigns against SARS-CoV-2 PL^Pro^ yielded two UbVs, UbVs.CV2.1.a and UbVs.CV2.1.b, that potently inhibited deISGlyation and the proteolytic activity of SARS-CoV-2 PL^Pro^, reducing viral replication, in cell culture, by almost five orders of magnitude [78].

#### 3.3.4. UbVs as Probes of DUB Activity

In addition to their potential in therapy development, UbVs have also been employed as activity-based probes (ABPs) for the identification of DUB-interacting proteins within complex cellular systems. Recently, Hewitt et al. developed UbV-ABPs to investigate the interactors of UCHL1, UCHL3, and UCHL5, members of the UCH family of DUBs. Critical to their approach was the in silico modeling and prediction of UbVs with single-point mutations that would enhance specificity and affinity for UCHL1 [79]. Many UbV-containing Thr9 mutations exhibited improved binding for UCHL1, with UbV^T9F^ showing a 35-fold greater specificity for UCHL1 over UCHL3 in vitro. A second T66K mutation was introduced to reduce USP-family cross-reactivity, as USP binding to Ub is susceptible to Thr66 mutations [80]. The resulting UbV (UbV^T9F/T66K^) served as a UCHL1-selective UbV-ABP, but it unexpectedly displayed preferential binding for UCHL3 over UCHL1 when used in small cell lung cancer cell lysates [79]. In a subsequent study, the same group focused on designing UCHL3-selective UbV-ABPs and developed UbV^Q40V/T66K/V70F^-ABP, which showed outstanding UCHL3 inhibition in vitro and reactivity for UCHL3 in MDA-MB-231 breast cancer cells without cross-reactivity with other DUBs [81].

Two additional UbV-ABPs were designed for USP7 and USP16. By combining USP7-specific Ub7Ub25.2540 with additional in silico predicted mutations, an improved UbV, named M6, was developed [82]. Biotinylated M6-ABP was successfully used as an affinity reagent for the enrichment of interactors of USP7, enabling the detection of two known USP7 partners (PPlL4 and DHX40) and two other USPs (USP15 and USP16) [82]. A similar approach yielded M20, a USP16-specific UbV used to develop another UbV-ABP [83]. Biotinylated M20-ABP proved to be an extremely selective probe of USP16 in affinity purification coupled to mass spectrometry (AP-MS) experiments. Additionally, when conjugated with an AMC (7-amido-4-methylcoumarin) moiety, M20 showed superb performance in detecting USP16 activity within cells [83]. These findings highlight the utility of UbVs as valuable tools for the specific and sensitive detection of DUB activity.

#### 3.3.5. UbVs Targeting Noncatalytic Protein Domains in the UPS

Targeting catalytic sites within UPS enzymes effectively modulates their activity; however, their conservation among enzyme family members limits selectivity and inhibitory potency, thereby increasing the risk of toxicity. In contrast, noncatalytic sites are generally less conserved and present an attractive option for expanding the druggable space within the UPS. In this regard, ubiquitin binding domains (UBDs) offer a unique opportunity as they contribute to Ub recognition without directly participating in catalytic activity.

Ubiquitin interacting motifs (UIMs) are short (~20 amino acid long) alpha-helical UBDs that recognize Ub with a weak affinity and in different conformations and orientations, adding to the versatility and functional diversity of enzymes in the UPS [84]. Using a naïve UbV phage display library, we isolated a UbV with nanomolar affinity for the first UIM of the yeast protein Vps27 (Figure 3b), moderate affinity for the third UIM of UFO1, and no binding to a panel of 10 other yeast UIMs [85,86]. Using this UbV as a scaffold, we constructed a UIM-focused UbV phage display library which we employed in selection campaigns against all human UIMs. Remarkably, UbVs were successfully engineered for 42 out of the 60 human UIMs present in 31 different proteins [87]. Despite the limited surface area of UIMs, a significant portion of the isolated UbVs displayed remarkable specificity. Among the identified UbVs, 52% of isolated UbVs recognized three or fewer UIMs, and 26% only recognized their cognate UIM [87]. These isolated UbVs were even able to discriminate between UIMs that shared >80% sequence identity, highlighting their high specificity. To investigate USP inhibition via UIM binding, we targeted the UIM of USP28 with a UbV and proved its effectiveness in blocking the hydrolysis of K11-linked di-Ub chains by USP28 in a dose-dependent manner. Thus, it is possible to obtain USP28 inhibitors by targeting domains beyond the catalytic site of the enzyme. In addition to its potency in vitro, the same UbV also demonstrated efficacy in live cells, reducing cellular levels of the USP28 substrate MYC, a pro-oncogenic transcription factor [87]. To further emphasize the functional role of UIMs in catalysis, UbVs specifically designed to target the three UIMs in USP37 successfully inhibited the processivity of K48-linked di-Ub chains by USP37 [88]. This inhibition not only demonstrated the functional role of UIMs in the catalytic activity of USP37 but also provided insight into the underlying mechanism of action of the enzyme.

These findings highlight the importance of UIMs in modulating the activity of USPs, shed light on their mechanisms of action, and provide an alternative strategy for targeting currently undruggable USPs. This strategy can be expanded to other UBDs such as the C-terminal ubiquitin-like domains (Ubls) of USP7, which play a crucial role in Ub recognition [89]. Although the development of small molecule inhibitors targeting UBDs is still limited, a notable example is the zinc-finger UBD of USP5 for which a single molecule inhibitor has been developed [90].

Targeting specific regulatory protein domains, such as the DUSP domain found exclusively in USPs, offers a more precise approach for the selective targeting of proteins within the UPS. The DUSP domain plays a crucial role in enhancing the activity of USP4 by Ub removal from its catalytic site [91]. Using a noninhibitory UbV specific for the DUSP domain of USP15 as a starting point [69] (Figure 3b), we constructed a novel DUSP-focused UbV phage display library [92]. To ensure binding specificity even among homologous DUSPs, we preincubated phage pools with noncognate DUSP proteins before performing affinity selections against the target cognate DUSPs. Through this approach, we successfully isolated UbVs that specifically recognize the DUSP domains of USP15, USP4, USP11, USP20, and USP33, discriminating even between DUSPs sharing >65% sequence identity. Remarkably, these UbVs demonstrated a potent inhibition of the catalytic activity of USP15, USP11, and USP20 through their interaction with the DUSP domain [92], suggesting that DUSPs can serve as alternative targets for the selective inhibition of currently undruggable USPs.

UbVs have also been used to discover novel regulatory surfaces of members of the UPS. Phage display enabled the isolation of a UbV targeting the APC2 subunit of the E3 ligase anaphase-promoting complex/cyclosome (APC/C) [93]. This UbV disrupts the interaction between the APC2-WHB domain and the E2 UBE2C, inhibiting UBE2C-mediated ubiquitination. Further structural studies revealed that the UbV bound to a previously undescribed Ub-binding site (Figure 3b), highlighting the potential of UbVs for inhibiting UPS proteins via interactions with novel surfaces [93].

Finally, UbV technology was also used to identify modulators of the UPS for enhancing genome engineering. Specifically, i53, a UbV targeting the Tudor ubiquitin-dependent recruitment (UDR) domain of 53BP1 (Figure 3b), disrupted the recognition of 53BP1 for methylated histone analogs, blocking its accumulation at DNA damage sites, thereby increasing the efficiency of precise genome editing [94,95].

These studies showcase the promising potential of targeting noncatalytic domains within the UPS, presenting a novel approach for modulating protein activities without directly targeting their catalytic function.

## 4. Targeting Non-UPS Proteins with UbVs

The unique biochemical and biophysical properties of Ub have prompted scientists to explore its potential as a naturally occurring, stable, and well-characterized scaffold for targeting proteins beyond the UPS.

To identify an optimized Ub scaffold, Leung et al. employed a UbV phage display library containing a N-terminal FLAG tag [23]. Panning of this library against an anti-FLAG tag antibody enabled the isolation of UbVs with correct folding and removal from the phage pool of unstable UbVs. In-depth analysis revealed that G10, K11, A46, G47, and Q49 residues were intolerant to mutations [23]. Notably, G10 mutations induce the formation of β-strand UbV dimers, as in the case of UbV.XR targeting the RING domain of XIAP [43] and UbV.D15 binding the DUSP domain of USP15 [69] (Figure 3b).

Based on these observations, two tailored UbV libraries were constructed and used to isolate UbVs binding to the Her3 extracellular receptor and a diverse array of intracellular modular domains characterized by different structures and functions. UbVs selected against Her3 showed binding to their cognate target on a site that is distinct from that of Her3 physiological ligand neuregulin [23]. Notably, these UbVs displayed high specificity, as they failed to bind other members of the epidermal growth factor receptor family. One UbV, named UbV.H3.2, demonstrated remarkable nanomolar affinity for Her3 and enabled specific and efficient immunoprecipitation of Her3 homo- and heterodimers from cells [23].

Affinity-selection campaigns against the phosphotyrosine-specific (pTyr) SH2 (Src homology 2) domain of the growth factor receptor-bound protein 2 (Grb2) adaptor protein yielded two specific UbVs [23]. While Ubv.G2.1 displayed an exclusive recognition of the Grb2 SH2 domain, Ubv.G2.2 recognized also the SH2 domains of GRAP and GADS, which share 63% and 56% identity with the Grb2 SH2 domain, respectively. Interestingly, Ubv.G2.1, unlike Ubv.G2.2, did not exhibit inhibitory effects on the binding of Grb2-SH2 to an immobilized pTyr-peptide, suggesting that Ubv.G2.1 does not interact with the pTyr-binding site of the SH2 domain. Genetically encoded fusion of these UbVs to improve their affinity via avidity effects enabled the immunoprecipitation of endogenous Grb2 and reduced Grb2-mediated signaling in the EGFR pathway, demonstrating that UbVs can be used to target proteins beyond the UPS [23].

While the engineering of a single-Ub chain successfully enabled the modulators of the UPS as well as binders for non-UPS proteins, the limited surface area of the Ub scaffold has prompted the exploration of diubiquitin-based scaffolds, such as Affilin [96], to expand the versatility and capabilities of Ub-based binders.

Lorey et al. developed a phage display Affilin library to isolate binders against the human fibronectin extra-domain B (ED-B) [96], a glycoprotein almost exclusively expressed in tumor tissues [97]. Through phage display selections and ribosome-display-mediated affinity maturation, they successfully isolated an Affilin variant with extraordinary picomolar affinity, outstanding selectivity, and excellent thermodynamic stability [96]. In tumor-bearing mice, the intravenous administration of the ED-B-specific Affilin demonstrated significantly enhanced localization compared to a Ub dimer control, without causing toxicity [96]. However, due to its small size (∼17 kDa), the ED-B-specific Affilin exhibited a short half-maximum serum clearance of 30 min. To extend its systemic circulation time, Lorey et al. leveraged the modularity of Affilins by fusing their lead candidate to an antibody Fc fragment, leading to the development of an Affilin variant with an extended half-life of 56 h, providing a strategy for prolonged circulation and tumor retention [96].

Thus, the unique biochemical and biophysical properties of Ub, combined with its low risk of potential immunogenicity, make it an excellent scaffold for generating novel therapeutics targeting proteins beyond the UPS.

## 5. UbV-Mediated Targeted Protein Degradation

In the past decade, significant efforts have been made for developing small molecules that recruit pathogenic proteins to the UPS for targeted protein degradation. These endeavors have given rise to innovative approaches such as PROTACs (proteolysis targeting chimeras) and molecular glues. PROTACs are heterobifunctional molecules consisting of an E3 ligase-recruiting ligand and a protein-targeting warhead, connected by a linker [14] (Figure 4). By recruiting the protein of interest to an E3 ligase, PROTACs facilitate its ubiquitination and subsequent degradation by the UPS. Similarly, molecular glues are monovalent small molecules that can bind both an E3 ligase and a protein of interest, bringing them into close proximity and promoting ubiquitination and degradation [98] (Figure 4). These approaches offer a unique advantage over traditional small molecule inhibitors by directly removing the target protein from the cellular environment, leading to more profound and long-lasting therapeutic effects.

Inspired by the success of these strategies, many groups have developed novel methods for the development of genetically encodable PROTACs, exemplified by peptide-based PROTACs and ubiquibodies, among others [99,100]. Indeed, UbVs have demonstrated their ability to engage the UPS, which has led to their application in the development of Ub variant-induced proximity (UbVIP) reagents (Figure 4 and Table 1) [101].

UbVIPs are linearly linked UbVs consisting of a protein-targeting UbV, such as i53 (that binds the UDR domain of 53BP1, described earlier in this review), and a noninhibitory UbV that recruits an E3 ligase [40]. A UbVIP obtained by the fusion of i53, via glycine–serine linkers with different lengths, to a UbV recruiting either RFWD3 or NEDD4L E3 ligases remarkably led to the near complete degradation of 53BP1 [101]. Similarly to PROTACs, the nature of the linker connecting the E3-engaging UbV and the UbV-targeting warhead had a significant effect on the activity of UbVIPs, with molecules containing an eight-residue long linker showing the highest efficacy [101,102].

This proof-of-concept study demonstrates the potential utility of previously identified UbVs in providing novel protein-based protein degraders.

## 6. Conclusions

Over the past decade, our repertoire of UbVs has expanded beyond targeting the catalytic sites of UPS proteins to include various UBDs and non-UPS proteins (Table 2). Various engineering techniques have been used to generate UbVs, such as rational design, phage display, ribosome display, and computational structure-based methods. Despite the limitations of a library with reduced sequence diversity, intracellular yeast-two-hybrid screening has proven valuable in identifying UbVs with enhanced functional potential in cell-based experiments [63]. Moreover, the development of novel UbV modalities, such as UbVs with extended loops, genetically fused UbV dimers, and β-strand-swapped UbV dimers, has expanded the surface area of UbVs [43,46,69,86,87], enabling interactions with noncanonical epitopes on target proteins.

These approaches have been applied to systematically target diverse protein families, including E2 ligases, different types of E3 ligases (HECT-E3 ligases, RING-E3 ligases, U-box E3 ligases, and the SCF complex), and various DUBs such as USPs, UCHs, and JAMMs.

Structural studies have revealed that UbVs can bind to canonical Ub-binding sites, including catalytic sites, and to allosteric hydrophobic sites that contribute to regulate enzyme activities. This unique property is attributed to the extended surface area of UbVs compared to small molecules, allowing for diverse and versatile interactions with target proteins. As a result, UbVs have not only facilitated the development of inhibitors for UPS proteins but have also enabled the discovery of enzyme activators [40,43], significantly expanding their potential applications.

Furthermore, UbVs targeting noncatalytic protein domains have demonstrated their ability to competitively block protein interactions, including those with Ub or natural binding partners [85,87,92]. This mechanism of action has expanded our mechanistic understanding of enzymes in the UPS and provided valuable insights into novel strategies for the pharmacological inhibition of enzymes that were previously considered undruggable. Therefore, UbVs offer a promising approach for modulating the activity of these enzymes and expanding the repertoire of potential therapeutic targets within the UPS.

Within cells, UbVs have successfully demonstrated their expected cellular effects, functioning as inhibitors or activators with therapeutic potential. Particularly noteworthy is their effectiveness as antiviral reagents, as they can effectively disrupt viral replication by targeting essential enzymes involved in polyprotein processing [76,78]. The high conservation rate of viral proteases compared to the surface-exposed proteins required for cell binding and entry [103] makes UbVs an optimal tool for preparedness campaigns against future emerging pandemics.

However, similar to peptides and other biologics, a key issue in the use of UbVs as therapeutics is their intracellular delivery for effective interaction with the UPS. The fusion of UbVs with cell-penetrating peptides (CPPs), such as TAT and Penetratin [104], provides a viable strategy to overcome the cell membrane barrier. However, the delivery efficiency of peptide/protein cargos using CPPs has been underwhelming, often requiring high concentrations that can contribute to cytotoxicity [105].

Alternatively, viral vectors offer an additional avenue for therapeutic molecule delivery, as they can be engineered to enhance target specificity and boost transgene expression [106,107]. However, despite recent advancements in their development, they pose safety concerns for in vivo applications. Nonetheless, recently, adeno-associated viruses (AAVs) were successfully employed for the delivery of UbV i53 into human and mouse cells. AAV-mediated delivery of i53 enabled quantitative expression of the UbV, resulting in efficient and precise genome editing by stimulating homology-dependent repair (HDR) [94].

In another example, UbV7.2 (targeting USP7) was delivered into mammalian cells via phage-like particles (PLPs), obtained through engineering of the bacteriophage λ capsid [108]. Fusion of UbV7.2 with the λ decoration protein (gpD) enabled the display of the UbV on the capsid surface [109]. Additionally, the decoration of PLPs with trastuzumab and fluorescein enabled the specific targeting of Her2-overexpressing cells and particle tracking. Using this strategy, PLP-UbV7.2 was internalized into SKBR3 cancer cells and accumulated in the endosomal compartment. Interestingly, UbV7.2 was slowly released by the endosomes, inhibiting USP7 and resulting in cancer cell apoptosis [109].

Another therapeutic opportunity for UbVs is highlighted by the studies of Affilins, as they have shown stability in the bloodstream and potential efficacy as extracellular receptor binders [96]. These efforts pave the way for the development of novel UbVs that can be administered intravenously for therapeutic purposes. The ability of UbVs to engage extracellular targets [23,96] opens up new avenues for the modulation of protein activities outside the UPS, expanding their potential applications in various diseases and therapeutic interventions.

Finally, the power of UbVs lie in their usage as probes of the UPS. Numerous structural and functional studies have provided valuable insights into the functions of enzymes in the UPS and enabled the identification of regulatory surfaces that can modulate protein function. This wealth of information can be used for the structure-based design of small molecules, as well as the development of high-throughput methods for the screening of chemical libraries. Such an approach will enable the development of both inhibitors and activators, particularly those targeting noncanonical sites. As activity has already been tested for UbVs, the effects of small molecules targeting the same surface as UbVs can be predicted, expanding the druggable space of small molecules and opening new opportunities for therapeutic interventions.

**Table 2 cells-12-02117-t002:** List of characterized UbVs clustered based on the protein family they target.

Family	Target Domain	Protein ID	UbV Isolation Method	UbV ID	Bound Site	Function	Reference
E2	Catalytic	Ube2k	Phage display	UbV.k.1	UBC	Inhibitor	[31]
	Catalytic	Ube2k	Phage display	UbV.k.2	UBC	Inhibitor	[31]
	Catalytic	Ube2D1	Phage display	UbV.D1.1	backside	Inhibitor	[33]
	Catalytic	Ube2V1	Phage display	UbV.V1.1	backside	Inhibitor	[33]
	Catalytic	Ube2G1	Phage display	UbV.G1.1	backside	Inhibitor	[33]
E3	HECT	WWP1	Phage display	UbV.P1.1	E2-site	Inhibitor	[40]
	HECT	WWP2	Phage display	UbV.P2.3	ND	Activator	[40]
	HECT	ITCH	Phage display	UbV.IT.2	E2-site	Inhibitor	[40]
	HECT	*S. cerevisiae* Rsp5	Phage display	UbV.R5.4	N-lobe exosite	Activator	[40]
	HECT	NEDD4	Phage display	UbV.N4.4	N-lobe exosite	Activator	[40]
	HECT	NEDD4	Phage display	UbV.N.2	ND	Activator	[41]
	HECT	NEDD4L	Phage display	UbV.NL.1	ND	Activator	[40]
	HECT	NEDD4L	Phage display	UbV.NL.3	N-lobe exosite	Inhibitor	[40]
	HECT	HACE1	Phage display	UbV.HA.3	ND	Inhibitor	[40]
	HECT	SMURF2	Phage display	UbV.S2.5	ND	Inhibitor	[40]
	RING	pCBL	Phage display	UbV.pCBL	E2-site	Inhibitor	[43]
	RING	XIAP	Phage display	UbV.XR	RING/donor Ub	Activator	[43]
	U-Box	UBE4B	Phage display	UbV.E4B	E2 site	Inhibitor	[43]
	RCL	SCF	Phage display	UbV.Fw7.5	F-box interface	Inhibitor	[46]
	RCL	SCF	Phage display	UbV.Fw11.2	F-box interface	Inhibitor	[46]
	RCL	SCF	Phage display	UbV.Fl11.1	ND	Inhibitor	[47]
	RCL	SCF	Phage display	UbV.L1.1	ND	Inhibitor	[47]
	RCL	SCF	Phage display	UbVO11.1	ND	Inhibitor	[47]
	RING	APC11	Phage display	UbV.R	E2-site	Inhibitor	[93]
	APC/C	APC2	Phage display	UbV.W	Ub exosite	Inhibitor	[93]
DUBs	Catalytic	USP2a	Computational/phage display/Y2H	UbV2.3 and UbV2.1	ND	Inhibitor	[41]
	OTU	OTUB1	Phage display	UbV.B1.1	Ub-distal site	Inhibitor	[41]
	JAMM	STAMBP	Phage display	UbV.SP.1	Ins-1	Inhibitor	[74]
	JAMM	STAMBP	Phage display	UbV.SP.3	ND	Inhibitor	[74]
	UCH	UCHL1	Computational/ rational design	UbV^T9F/T66K^	ND	Inhibitor/ABP	[79]
	UCH	UCHL3	Computational/ rational design	UbV^Q40V/T66K/V70F^	ND	Inhibitor/ABP	[81]
	Catalytic	USP8	Phage display	UbV.8.2	Ub-binding site	Inhibitor	[41]
	Catalytic	USP21	Phage display	UbV21.4	ND	Inhibitor	[41]
	Catalytic	USP7	Phage display	Ub7Ub25	ND	Inhibitor	[59]
	Catalytic	USP7	Phage display	Ub7Ub25.2540	ND	Inhibitor	[59]
	Catalytic	USP7	Computational/phage display	UbV.7.2 and M6	Ub-binding site	Inhibitor/ABP	[60,82]
	Catalytic	USP10	Phage display	UbV10.1	ND	Inhibitor	[60]
	Catalytic	USP2	Y2H	UbV2.6	Ub-binding site	Inhibitor	[63]
	Catalytic	USP15	Phage display	UbV.15.1	Ub-binding site	Inhibitor	[69]
	Catalytic	USP15	Phage display	UbV.15.1/D (dimer)	Ub-binding site and DUSP	Inhibitor	[69]
	Catalytic	USP16	Computational	M20	ND	Inhibitor/ABP	[83]
	Catalytic	USP14	Phage display	U14Ub14	Ub-binding site	Inhibitor	[110]


vDUBs	OTU	CCHFV	Phage display	UbV.CC.4	Ub-binding site	Inhibitor	[76]
	PL^pro^	MERS-CoV	Computational/phage display	UbV.ME.4	Ub-binding site	Inhibitor	[76]
	PL^pro^	SARS-CoV-2	Phage display	UbV.CV2.1a	Ub-binding site	Inhibitor	[78]
UBDs	UIM	*S. cerevisiae* Vsp27	Phage display	UbV.v27.1	Ub-binding site	Inhibitor	[85,86]
	UIM	ANKRD13D	Phage display	UbV.ANKRD13D.4	Ub-binding site	Binding	[87]
	UIM	USP28	Phage display	UbV.USP28	ND	Inhibitor	[87]
	UIM	*D. rerio* USP37	Phage display	UbV.UIM	ND	Inhibitor	[88]
	DUSP	USP11	Phage display	UbV.11D.2	ND	Inhibitor	[92]
	DUSP	USP15	Phage display	UbV.15D.1	DUSP	Inhibitor	[92]
	DUSP	USP20	Phage display	UbV.20D2	ND	Inhibitor	[92]
Adaptor protein	SH2	Grb2	Phage display	UbV.G2.1	ND	Binding	[23]
	SH2	Grb2	Phage display	UbV.G2.2	pTyr-binding site	Inhibitor	[23]
	SH2	Grb2	Phage display	UbV.G2.2/1 (dimer)	pTyr-binding site and SH2	Inhibitor	[23]
EGFR	Ecto-domain	Her3	Phage display	UbV.H3.2	Diverse from neuregulin site	Binding	[23]
Fibronectin	ED-B	Oncofetal fibronectin	Phage/ribosome display	Affilin-77405	ND	Binding	[96]
Tumor suppressor	UDR	53BP1	Phage display	i53	UDR domain	Gene editing/ targeted degradation	[94,95,101]

The target domain and the method used to isolate UbVs are indicated, as well as their binding site (bound site) and reported function. ND: not determined. APC/C: anaphase-promoting complex; DUSP: domain present in ubiquitin-specific protease; DUBs: deubiquitinating enzymes; EGFR: epidermal growth factor receptor; Grb2: growth factor receptor-bound protein 2; HECT: homologous to the E6-AP carboxyl terminus; JAMM: JAB1/MPN/MOV34 metalloprotease; OTU: ovarian tumor protease; pCbl: phosphorylated Cbl; PL^pro^: papain-like protease; pTyr: phosphotyrosine; RCL: RING-E3 complex; RING: really interesting new gene; SH2: Src homology 2; UBD: ubiquitin-binding domain; UCH: ubiquitin C-terminal hydrolase; UIM: ubiquitin-interacting motif; UDR: Tudor ubiquitin-dependent recruitment domain; USP: ubiquitin-specific protease; vDUBs: viral deubiquitinating enzymes.

## 7. Future Challenges in UbV Technology

Despite their success, UbVs face challenges that must be addressed for their widespread use as tools to understand the UPS and as innovative therapeutics in various diseases:Specificity and selectivity: Ensuring high specificity and selectivity for target proteins is crucial for therapeutic applications of UbVs. As the number of identified UbVs increases, it becomes essential to address potential off-target effects and enhance the ability to discriminate between homologous proteins.UbV discovery strategies: While in vitro affinity selection has yielded successful protein-based inhibitors or activators of the UPS, some UbVs, like those targeting USP2, may lack functional activity within the cellular context. Exploring alternative strategies for isolating intracellularly functional UbVs can expedite discovery and enhance their therapeutic and research potential.Modulating proteasome activity: Small molecules like bortezomib and carfilzomib have demonstrated efficacy in inhibiting proteasome activity, leading to their approval for the treatment of certain cancers. Similarly, UbVs targeting specific proteasome subunits or regulatory proteins could offer a novel approach for modulating proteasome function with potentially greater selectivity and fewer off-target effects compared to traditional small molecules.Intracellular delivery: Efficient intracellular delivery of UbVs is essential for their therapeutic potential. Overcoming cellular barriers requires the development of novel delivery strategies such as novel CPPs, capable of escaping endosomes. Additionally, safer AAVs or lipid nanoparticles encapsulating the mRNA-encoding UbVs could unlock their translational potential.In vivo models: Establishing robust in vivo models that accurately represent the pathogenic activity of the UPS will be crucial for validating the therapeutic efficacy of UbVs.Addressing these challenges will unlock the full translational potential of UbVs and contribute to advancing our understanding of the UPS, leading to new opportunities for therapeutic interventions in various diseases.

## Figures and Tables

**Figure 1 cells-12-02117-f001:**
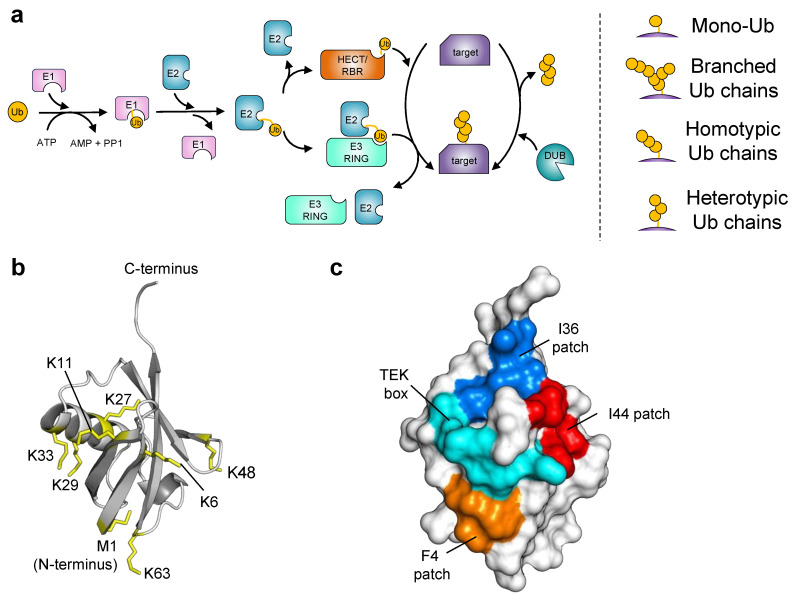
Key steps in the ubiquitination reaction and features of ubiquitin structure. (**a**) Schematic of the ubiquitination and deubiquitination reactions. Ubiquitination is a multistep cascade of enzymatic reactions involving the activation of Ub by the E1 Ub-activating enzyme followed by its transfer to an E2 Ub-conjugating enzyme. Subsequently, Ub is transferred from the Ub~E2 conjugate to a specific lysine residue in the substrate protein, either directly by a RING E3 Ub ligase or through a HECT/RBR E3 Ub ligase. Ubiquitin itself can act as a substrate for the conjugation of Ub molecules, leading to the generation of complex polyubiquitin chains (right panel). The removal of Ub modifications from substrate proteins is mediated by the activity of deubiquitinases (DUBs). (**b**) Structure of Ub (PDB: 1UBQ) highlighting the seven lysine residues and methionine involved in polyubiquitin linkages shown as yellow sticks. (**c**) Surface representation of Ub-interacting surfaces. Ub is shown as a gray surface, with amino acid patches involved in the recognition of UPS members colored in red (Ile44 patch: Leu8, Ile44, His68, and Val70), blue (Ile36 patch: Leu8, Ile36, Leu71, and Leu73), cyan (TEK box: Lys6, Lys11, Thr12, Thr14, and Glu34), and orange (F4 patch: Gln2, Phe4, and Thr14).

**Figure 2 cells-12-02117-f002:**
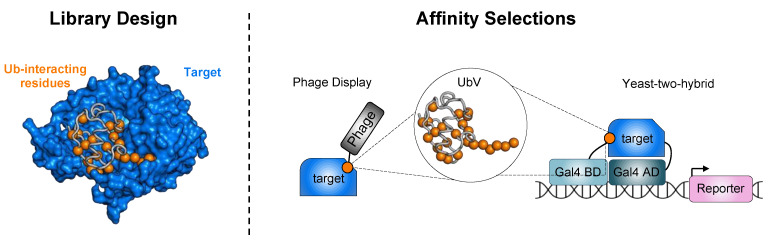
Design and selection of high-affinity UbVs. (**Left** panel) Structural analysis of Ub in complex with UPS proteins enables the identification of key Ub residues involved in binding. Here, as an example, we show the structure of Ub in complex with USP2a (target, blue) (PBD: 3NHE). The Ub backbone is displayed as a gray tube with interacting residues shown as orange spheres. (**Right** panel) Affinity selections using phage display and yeast-two-hybrid libraries are performed to identify UbVs with enhanced affinity to protein targets. In the case of phage display, the target protein is immobilized onto a solid support (such as microtiter plates or beads) and incubated with UbVs displayed on the phage surface. Following three to five rounds of affinity selection, specific individual phage-displayed UbVs are selected and analyzed. In yeast-two-hybrid systems, UbV libraries and a target protein acting as either a bait or prey protein are fused to either a Gal4-binding domain (Gal4 BD) or a Gal4-activation domain (Gal4 AD). The interaction of the UbV and target protein leads to the activation and transcription of a downstream reporter allowing for the detection of binding events.

**Figure 3 cells-12-02117-f003:**
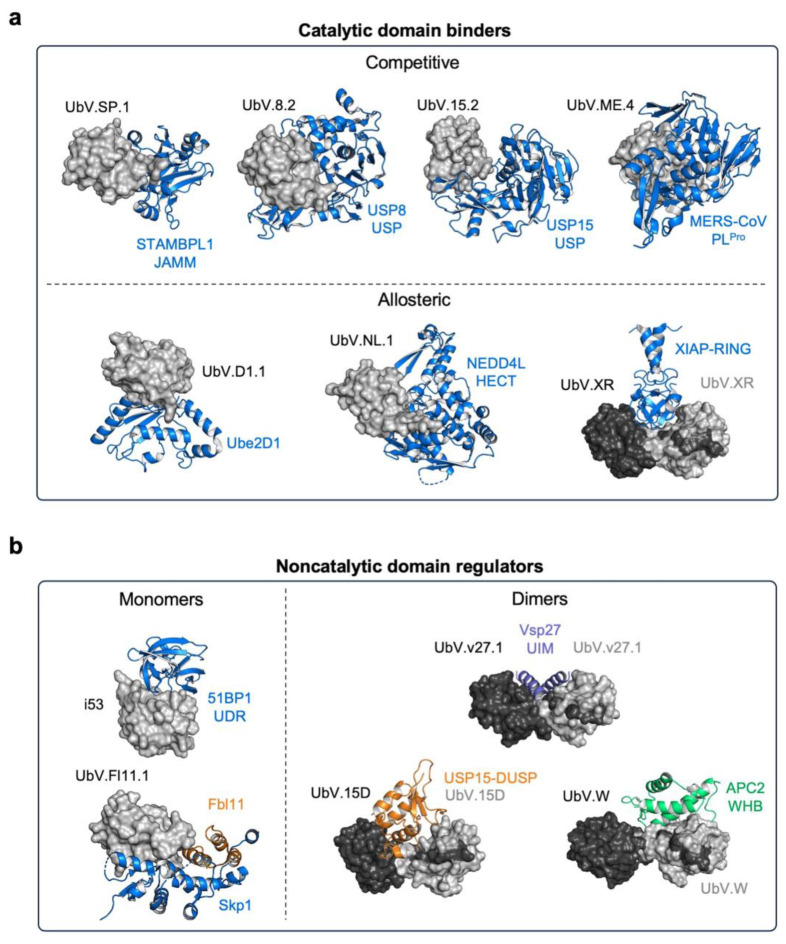
Structural analysis of UPS members in complex with UbV modulators. (**a**) UbVs regulating the catalytic activity of enzymes in the UPS can either directly compete with Ub binding (top panel, competitive) or modulate enzyme activity allosterically (bottom panel, allosteric). UbVs are depicted as a gray surface, whereas their interacting partners are represented as blue cartoons. Among the shown competitive UbV inhibitors are UbV.SP.1 and the STAMBP-JAMM domain (PDB: 7L97), UbV.8.2 bound to the USP8-USP domain (PDB: 3N3K), UbV.15.2 in complex with the USP15-USP domain (PDB: 6CRN), and UbV.ME.4 bound to MERS-CoV PL^pro^ (PDB: 5V69). Examples of allosteric UbV modulators are UbV.D1.1 in complex with E2 ligase Ube2D (PDB: 6D4P) and UbV.NL.1 bound to the HECT domain of NEDD4L E3 ligase (PDB: 5HPK). UbV.XR forms a homodimer in which each UbV monomer (black and gray surfaces) interacts with the RING domain of the E3 ligase XIAP (blue, PDB: 5O6T). (**b**) UbVs regulating protein activity through binding to noncatalytic protein domains include UbV monomers (gray), as in the case of i53 bound to the UDR domain of 53BP1 (blue, PDB: 5J26) and UbV.Fl11.1 in complex with Fbl11 (orange) and Skp1 (blue, PDB: 6BYH). UbVs can also assemble into symmetric dimers engaging their partners in a 2:2 binding stoichiometry, such as in the case of UbV.v27.1 bound to the UIM domain of Vsp27 (purple, PDB: 6NJG). However, UbVs can also form asymmetric dimers interacting in a 2:1 binding stoichiometry with their partners. Examples of asymmetric UbV dimers are UbV.W bound to the WHB domain of APC2 (green, PDB: 6OB1) and UbV.15D in complex with the DUSP domain of USP15 (orange, PDB: 6DJ9).

**Figure 4 cells-12-02117-f004:**
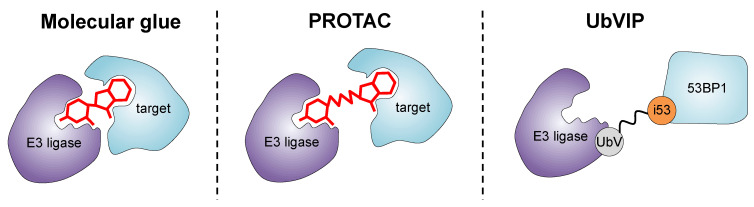
Strategies for inducing targeted protein degradation. Molecular glues are monovalent molecules that induce the dimerization of an E3 ligase with a protein target. Unlike molecular glues, PROTACs are heterobifunctional degraders that simultaneously bind both an E3 ligase and the protein of interest to be degraded. UbVIPs are genetically encoded degraders, obtained by fusing i53 (a UbV targeting the UDR domain of 53BP1) via a flexible linker to a UbV binding an E3 ligase.

**Table 1 cells-12-02117-t001:** Comparison of the features of small molecule degraders (PROTACs and molecular glues) and UbV-dependent targeted degradation (UbVIPs).

Features	Small Molecule Degraders	UbVIPs
Target Protein Range	Broad spectrum	Selective targeting of specific proteins within the UPS and beyond
Mechanism of Action	Recruit target protein to E3 ligase for ubiquitination	Facilitate ubiquitination and degradation of target proteins. Can enhance E3 ligase ubiquitination activity
Delivery and Cellular Uptake	Diffusion across cell membranes	Require specialized methods
Interaction with the UPS	Engage only a small subset of E3 ligases	Recruit diverse E3 ligases with distinct activity and cellular localization. Can engage E2 enzymes and DUBs
Target Protein Degradation Speed	Rapid and reversible	Can vary depending on UbV design and abundance
Cellular Response	Transient degradation of target protein	Transient modulation of target protein degradation
Binding Site Specificity	Depend on ligand and target protein interaction	Highly specific due to modular customization of UbVs
Protein Degradation Scope	Limited by the availability of specific ligands	Flexible and can be applied to a wide range of target proteins
Druggability	Require development of specific ligands for targets	Expand the repertoire of druggable targets even beyond the UPS

DUBs: deubiquitinating enzymes; UPS: ubiquitin–proteasome system.

## Data Availability

No new data were created or analyzed in this study. Data sharing is not applicable to this article.

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
