# Peer review of "Ubiquitin Engineering for Interrogating the Ubiquitin–Proteasome System and Novel Therapeutic Strategies"

_cells, 2023, doi:10.3390/cells12162117_

Round 1

Reviewer 1 Report

The present review by Tang et al. thoroughly covers the use of ubiquitin variants as tools to modulate ubiquitin-mediated proteostasis.  The authors are experts in the topic, especially in the use of ubiquitin variants for targeting ubiquitin-interacting motifs (UIMs), to modulate the reading of and fates arising from ubiquitin-UIM interactions.  The review is comprehensive and up-to-date, probably covering all publications in which ubiquitin variants were designed and/or applied to biological systems.  It is well-written and organized in a logical manner.  There is good description of structural aspects to explain how a small protein like ubiquitin has several binding faces and that small alterations can lead to specific effects on the Ub cycle enzymes (E2s, E3s, DUBs).  The review is an important contribution the literature and should be published with minor revision.

Some specific points:

While the itemization of all reported UbVs and respective targets is useful, reviewer was hoping for more description about how UbV variants are designed and screened.  This is largely limited to lines 110-114 with sporadic mentions of Y2H and combinatorial libraries.  Also there is some mention of computational design.  Reviewer thinks that maybe 1-2 stand-alone paragraphs on guidelines and strategies to make and screen libraries for a particular target could make the review even better.  The authors are well-situated to provide such information.

Figure 2 legend (line 154-165) is lacking “Figure 2” written as text, as is the case for Figure 1 (line 87).

In Figure 2B, Figure 2B legend, and also in text (line 461), there are several mentions of “51BP1”… shouldn’t this be “53BP1”? (53BP1 is correct in Table 1, and in text line 540 and ref 62).

In some cases, reviewer detected that in-text references for particular points seemingly did not correspond to linked reference articles.  One example: lines 77-81 are contrasting other scaffolds to block UPS compared to ubiquitin variants, but citations 14 and 15 refer to reviews on NEDD4 and Liddle Syndrome.  Was this intended?  Reviewer would encourage authors to go through all in-text citations, matching with corresponding reference articles to confirm that the correct associations are made.

Reviewer 2 Report

The review article entitled ‘Ubiquitin Engineering for Interrogating the Ubiquitin-Proteasome System and Novel Therapeutic Strategies’ is very intriguing, well written, and embodies latest advancements in the field. Especially, the description of ubiquitination and the novel therapeutic strategies. The author could improve the review article.

1.     The mechanism of Ubiquitination can be illustrated in a picture.

2.     The author briefly mentioned about E3 ligases. The authors can add TRAF family members as there is amounting evidence about the cross talk between TGFbeta receptors and TRAF6.

References

TRAF6 as a potential target in advanced breast cancer: a systematic review, meta-analysis, and bioinformatics validation. Feier Zeng, Scientific Reports.

TRAF6 Stimulates the Tumor-Promoting Effects of TGFβ Type I Receptor Through Polyubiquitination and Activation of Presenilin 1. Science Signaling.

The type I TGF-beta receptor engages TRAF6 to activate TAK1 in a receptor kinase-independent manner.

TRAF6 ubiquitinates TGFβ type I receptor to promote its cleavage and nuclear translocation in cancer. 

3.     There are other types of ubiquitination’s like mono Ub, and multi mono Ub, please include along with other lysine-based ubiquitination as decribed in the article. 

4.     The authors included PROTAC mediated target protein degradation, a widely studied mechanism in the field of ubiquitination, Protein degraders picture would be a better addition for this section.

5.     The section about DUB’s is interesting, elaboration of DUB’s role and its potential therapeutic interventions need to be added.

6.     The authors can add a separate section and include the current clinical trials that are ongoing in the field.

Reviewer 3 Report

Ubiquitin is a small protein that plays a crucial role in protein degradation and regulation within cells. It acts as a "molecular tag" by attaching itself to other proteins, marking them for degradation or modifying their functions. In recent years, researchers have identified several ubiquitin variants or modifications that go beyond the conventional roles of ubiquitin. Ubiquitin variants have diverse roles in cellular processes, including protein degradation, DNA repair, cell signaling, and autophagy. Each variant typically targets a specific subset of proteins or performs distinct functions within the cell. Ubiquitin modifications often regulate protein stability, activity, and localization. Some ubiquitin variants can be conjugated to the same protein, leading to fine-tuned regulation of its function. However, determining the three-dimensional structures of ubiquitin variants and their interacting partners is crucial to understand their functional mechanisms and designing targeted therapies.

The scientific aspect of the current review article is valuable in the field of ubiquitin biology. The authors here managed to summarize the key studies using the ubiquitin variants library through text mining and referencing. Importantly the table “list of characterized UbVs clustered based on the protein family” is an excellent addition to this manuscript. Overall the manuscript is informative in nature and a rich source of recent research done in the field.

However, the manuscript lacks certain aspects. In the following points, I offer some suggestions.

1) The authors may generate a table based on the potential advantages and disadvantages based on current research of using Ubiquitin Variants, PROTACs, or molecular glue in targeted protein degradation. That could be an excellent addition to reflect why the systematic development of ubiquitin variants could bring a promising therapeutic modality in the future.

2) A study demonstrated, UbV displayed appreciable binding and inhibition selectivity for UCHL1, a DUB which expresses in the central nervous system under normal physiological conditions, over close structural homolog UCHL3 (PMID: 32865982). The selectivity of UbVs over different homologs is an important aspect. Here authors could consider adding some more related research that studied the UbVs targeted selectivity and specificity.

3) Small chemicals like bortezomib, and carfilzomib inhibit proteasome activity. Authors could consider adding some possible aspects or insight into modulating proteasome activity using UbVs based on current research. Also, this aspect could be added as a future challenge. Considering this authors may add a small section on “the future challenges in this emerging area of research” in a pointwise manner.
